# Trimethylamine *N*-Oxide (TMAO) and Indoxyl Sulfate Concentrations in Patients with Alcohol Use Disorder

**DOI:** 10.3390/nu14193964

**Published:** 2022-09-24

**Authors:** Laurent Coulbault, Alice Laniepce, Shailendra Segobin, Céline Boudehent, Nicolas Cabé, Anne Lise Pitel

**Affiliations:** 1Physiopathology and Imaging of Neurological Disorder (PhIND), Cyceron, Institut National de la Santé et de la Recherche Biomédicale (INSERM), Institut Blood and Brain @ Caen Normandie, Normandie University, Université de Caen (UNICAEN), 14000 Caen, France; 2Biochemistry Department, Caen University Hospital, 14000 Caen, France; 3Centre de Recherche sur les Fonctionnements et Dysfonctionnements psychologiques (CRFDP, EA7475), Normandie University, Université de Rouen (UNIROUEN), 76000 Rouen, France; 4Addictology Department, Caen University Hospital, 14000 Caen, France

**Keywords:** trimethylamine *N*-oxide, indoxyl sulfate, alcohol, alcohol used disorder, liver, cognition

## Abstract

Background: Trimethylamine *N*-oxide (TMAO) and indoxyl sulfate (IS) are produced by the microbiota and the liver, and can contribute to brain aging and impaired cognitive function. This study aims to examine serum TMAO and IS concentrations in patients with alcohol-use disorder (AUD) at the entry for alcohol withdrawal, and the relationships with several biological, neuropsychological, and clinical parameters. Methods: TMAO and IS were quantified in thirty AUD inpatients and fifteen healthy controls (HC). The severities of AUD and alcohol withdrawal syndrome (AWS), and general cognitive abilities were assessed in AUD patients. Results: TMAO concentrations did not differ between HC and AUD patients. Several biomarkers assessing nutritional status and liver function were significantly different in AUD patients with the lowest TMAO concentrations compared to other AUD patients. IS concentration was significantly lower in AUD patients and a significant positive predictor of serum prealbumin variation during the acute phase of alcohol withdrawal. No relationship was observed between the concentrations of these metabolites and the severities of alcohol dependence, AWS, or cognitive deficits. Conclusions: Our data suggest that AUD patients with low concentrations of TMAO or IS should probably benefit from a personalized refeeding program during the acute phase of alcohol withdrawal.

## 1. Introduction

Chronic and heavy alcohol consumption in patients with alcohol-use disorder (AUD) induces biological perturbations which contribute to liver disease, cancer and brain abnormalities [1,2]. Ethanol has its own effects on the body but is also metabolized in acetaldehyde in tissues, which is a reactive compound producing deleterious effects particularly on proteins and DNA [1,2].

AUD promotes an alteration of the gut barrier and increases intestinal permeability, contributing to modify the absorption of micronutrients. It contributes also to the transfer of various metabolic compounds produced by the gut microbiota in the portal veinous bloodstream, such as lipopolysaccharide (LPS) or indol compounds [3], and to bacterial translocation. AUD can also promote gut dysbiosis characterized by an alteration of the microbiota composition [3]. All together, these mechanisms facilitate the development of a low grade inflammation, liver disease and probably play a role in brain neuroinflammation in AUD patients [4].

Trimethylamine (TMA) is a compound produced by the gut microbiota and come from the metabolism of choline, betaine and L-carnitine, three compounds found in food, mainly from animal origin [5]. TMA is absorbed in the gut and is rapidly metabolized in the liver by flavin containing monoxygenase 1 and 3 (FMO1 and FMO3) to produce trimethylamine-*N*-oxide (TMAO). Recent studies showed that this compound may play a role in various diseases. TMAO is found, for example, at high levels in renal failure, and composition of diet seems to affect its blood concentrations [6,7,8,9]. However, the effect of diet on TMAO plasma concentration is still a matter of debate. Although TMAO concentration can vary in a large extent after carnitine supplementation [8], a recent study showed that TMAO plasma concentrations are not affected by consumption of eggs (a food rich in choline) in healthy volunteers [7,10,11,12]. Another study in a population of older women showed that a high protein diet can increase serum levels of carnitine and indoxyl sulfate (IS), but not TMAO serum concentrations [13]. 

In experimental models and human studies, high TMAO concentrations induce synthesis of proinflammatory cytokines, oxidative stress, endothelial dysfunction, vascular inflammation and perturbation of lipid metabolism (see Janeiro et al. [14] for review). These mechanisms contribute probably to insulin resistance and atherosclerosis, then promoting the development of cardiovascular diseases [14].

Many authors proposed that high concentrations of TMAO could play a role in the development of neurological disorders. TMAO is found in the cerebrospinal fluid, and could promote blood–brain barrier disruption, aggregation of Aβ protein and tau-tubulin assembly [14]. TMAO could also be involved in the perturbation of beta-amyloid and tau homeostasis and thus may play a role in the development of cognitive deficits in AUD patients [15,16,17]. Lastly, another recent experimental work showed that TMAO can contribute to brain aging [18]. Therefore, we can hypothesize that high TMAO concentrations contribute to cognitive deficits observed in AUD patients.

IS is another compound coming from the metabolic transformation of dietary tryptophan by the gut microbiota and the liver. Tryptophan, which is an essential amino acid, is metabolized in the gut to form various indole compounds, which go across the gut barrier and are then sulfoconjugated by the liver, and finally eliminated as IS by kidney. IS is considered like an uremic toxin as blood levels of this compound greatly increase during chronic renal failure and contribute to the development of cardiovascular diseases [19]. Recent experimental works also showed that IS can induce blood–brain barrier permeability, neuroinflammation [20,21], anxiety and depression-like behaviour in mice treated with IS [22]. In addition, IS administration in mice can alter cognitive performance [22,23,24].

There is a growing interest for TMAO and IS in cardiovascular and neurological diseases, and to our knowledge, these metabolites have not been studied yet in AUD patients. The objective of the present study was therefore to quantify serum TMAO and IS concentrations in a sample of AUD inpatients at treatment entry for alcohol withdrawal, and to examine their relationships with several biological markers, the severity of AUD, the severity of alcohol withdrawal syndrome (AWS), and general cognitive abilities evaluated at the end of the withdrawal.

## 2. Materials and Methods

### 2.1. AUD Patients and Healthy Controls

This study was approved by the local ethics committee (Comité de Protection des Personnes CPP Nord-ouest III; ALCOBRAIN study, NCT01455207). Thirty AUD patients were recruited during their alcohol withdrawal in the Addiction department of Caen University Hospital. They met the “alcohol dependence” criteria according to the DSM-IV-TR (American Psychiatric Association, 2000) or “severe AUD” criteria according to the DSM-5. To be included, participants had to have French as their native language, to be between 18 and 70-year-old. Comorbid conditions such as psychiatric disorders, a history of serious chronic disease, or neurological pathologies were exclusion criteria. All participants did not fulfill the criteria for substance use disorder (other than alcohol for AUD patients) except tobacco.

Fifteen healthy controls (HC) were also recruited to match the demographics (age, sex, education) of the AUD patients. All patients and HC had given a written and informed consent for blood sampling and analysis.

Fasting blood sampling was performed at treatment entry in the addiction department. Most AUD patients (*n* = 27) underwent a second blood sampling to measure nutritional markers such as serum albumin and prealbumin at the end of the acute alcohol withdrawal phase. Fasting blood sampling was performed at the inclusion for HC. Serum samples were stored at −80 °C up to analysis.

### 2.2. Alcohol Screening Tool and Assessment of Alcohol Withdrawal Syndrome (AWS)

The Alcohol Used Disorders Identification Test (AUDIT) was performed at the inclusion to identify the severity of alcohol harmful drinking and dependence in AUD patients [25]. HC were also interviewed with AUDIT [25] to ensure that they did not meet the criteria for DSM-IV-TR criteria for alcohol abuse (AUDIT < 7 for men and <6 for women).

In AUD patients, the severity of AWS was assessed using the Cushman’s maximum score [26]. This score takes the following clinical variables into account: hearth rate, systolic blood pressure, respiratory rate, tremor, sweating, agitation, and sensorial disorders. Each variable is scored from 0 to 3 according to the severity of the symptom. The Cushman score refers to the sum of these subscores.

### 2.3. Neuropsychological Examination

AUD patients underwent the Montreal Cognitive Assessment (MoCA) and Brief Evaluation of Alcohol-Related Neuropsychological Impairments (BEARNI). MoCA and BEARNI are two screening tools that have been validated in AUD patients to early detect neuropsychological impairments [27,28]. These tests were performed at the end of the acute withdrawal phase for AUD patients, and at inclusion in the study for HC.

### 2.4. Biological Analyzes

Biochemical and haematological analyzes were performed in the Biochemistry and Haematology departments of Caen University hospital. To explore a potential liver disease, the index of fibrosis Hepascore was determined using age, sex and serum concentrations of hyaluronic acid, α2-macroglobulin, bilirubin, and gamma glutamintranspeptidase (γGT), as previously described [29].

### 2.5. TMAO and IS Analysis

TMAO and IS quantitative analysis were performed using liquid chromatography coupled with tandem mass spectrometry (LC-MS/MS). Briefly, serum samples were mixed with deuterated internal standard (TMAO-d_9_ or IS-d_4,_ Sigma-Aldrich, Saint-Louis, MO, USA) and proteins were precipitated using cold methanol/acetonitrile (50/50 *v/v*). After centrifugation and dilution, supernatants were injected in an UFLC chromatographic system (Shimadzu, Kyoto, Japan) connected to a SCIEX QTRAP^®^ 5500 mass spectrometer (SCIEX, Toronto, ON, Canada). Chromatographic separation was performed using a Pursuit^®^pentafluorophenyl (PFP) column (Agilent technologies, Santa Clara, CA, USA) in gradient conditions, and using electrospray source in positive (TMAO) or negative (IS) mode. Multiple Reaction Monitoring (MRM) transitions were *m/z* 76→58 and m/z 85→66 for TMAO and TMAO-d_9_, respectively, and *m/z* 212→132 and *m/z* 216→136 for IS and IS-d_4_, respectively. All coefficients of variation (CV) were lower than 5% in validation assays. TMAO and IS determination was performed in all AUD patients and HCat the inclusion in the study, and only in 12 AUD patients at the end of alcohol withdrawal due to low sample quantity.

### 2.6. Statistical Analysis

This study was retrospective, as an ancillary investigation of large research protocol, and biological variability of IS or TMAO was not known in our population at the start of the study. It was then not possible to perform power calculation and to determine the number of subjects a priori.

All data were expressed as mean ± standard deviation. Statistical analysis was performed using Statistica software (Tibco software Inc., Palo Alto, CA, USA) and figures were made using GraphPad Prism software (Graphpad, La Jolla, CA, USA). Normality was checked using Kolmogorov–Smirnov tests. When data were not normally distributed, the variables were Log transformed to approach a normal distribution. Comparisons between AUD patients and HC were conducted using Student t test or ANOVA for comparison between tertiles of AUD patients (Appendix A). A paired sample t test was used to compare TMAO and IS variations of concentration during alcohol withdrawal (Appendix A). Differences were considered statistically significant at *p* < 0.05. Demographics, clinical neuropsychological and biological data are summarized in Table 1.

In AUD patients, we also performed simple Pearson correlations between TMAO or IS concentration on the one hand, and demographic, biological or clinical parameters on the other hand. Backward multiple regressions were then used to identify (1) the best predictors of TMAO or IS levels, and (2) whether TMAO or IS levels could be good predictors of different clinical, biological or neuropsychological parameters. Only the variables presenting a statistically significant correlation were entered in the regression model, and only predictors of the final solution are presented.

## 3. Results

### 3.1. Serum TMAO Concentrations in AUD Patients and HC

TMAO concentrations were not significantly different between AUD patients and HC (12.1 ± 19.2 µmol/L vs. 4.6 ± 2.5 µmol/L, NS; Table 1 and Figure 1A). However, an important inter-individual variability was observed for serum TMAO concentrations in AUD patients at treatment entry (Figure 1A). Some AUD patients had very low TMAO serum concentrations (<2 µmol/L) compared to HC, while others presented very high concentrations (>20 µmol/L, Figure 1A). Such very high or very low concentrations of serum TMAO were not observed in HC. We then explored whether TMAO concentrations could be related to biological, neuropsychological or clinical characteristics in AUD patients. In a first attempt, the biological and clinical profiles of AUD patients were studied according to their distribution profile into tertiles of serum TMAO concentrations (tertile 1, [TMAO] < 2.55 µmol/L; tertile 2, 2.55 µmol/L ≤ [TMAO] ≤ 6.76 µmol/L; tertile 3, [TMAO] > 6.76 µmol/L]; see Appendix A). Interestingly, AUD patients with the lowest TMAO values presented several significant biological alterations (see Appendix A). AUD Patients with higher TMAO concentrations presented the weakest biological perturbations compared to other AUD patients (see Appendix A).

### 3.2. Serum IS Concentrations in AUD Patients and HC

Serum IS concentrations were significantly lower in AUD patients when compared to HC (2.1 ± 1.0 µmol/L vs. 3.9 ± 1.5 µmol/L, *p* < 0.05; Table 1 and Figure 1B). The inter-individual variability in AUD patients and in HCwas lower by comparison to TMAO data.

The biological and clinical profiles of AUD patients has also been studied according to their distribution profile into tertiles of serum IS concentrations (tertile 1, [IS] < 1,7 µmol/L; tertile 2, 1.7 µmol/L ≤ [IS] ≤2.4 µmol/L; tertile 3, [IS] > 2.4 µmol/L]; see Appendix A).

Conversely to TMAO, no difference was seen when we compared tertiles of IS concentrations in AUD patients. However, the ASAT/ALAT ratio reached significance for the tertile 1 of AUD patients with the lower IS concentrations (see Appendix A). No correlation was found between TMAO and IS serum concentrations neither in AUD patients nor in HC (data not shown).

### 3.3. Evolution of Serum TMAO and IS Concentrations in AUD Patients during the Acute Alcohol Withdrawal Phase

TMAO and IS determination was performed only in 12 AUD patients at the end of the acute phase of alcohol withdrawal. Concerning TMAO concentrations, no significant difference was observed in these 12 AUD patients during before and after alcohol withdrawal. However, a high intra-individual variability was observed in these patients. Indeed, some AUD patients presented very high serum TMAO concentrations at treatment entry which typically decreased and returned at the end of alcohol withdrawal to serum levels usually observed in HC. However, it was not observed for all AUD patients, and in some cases, TMAO concentrations were quite similar at the end of alcohol withdrawal when compared to the concentrations at treatment entry (see Appendix A).

Concerning IS, serum concentrations significantly increased during the acute phase of alcohol withdrawal by applying a paired sample t test (*p* = 0.037). In addition, we observed a lower intra-individual variability when compared to TMAO concentrations (see Appendix A).

### 3.4. Correlations of Serum TMAO or IS Concentrations with Biological and Clinical Markers in AUD Patients

Simple correlations were performed in AUD patients between TMAO or IS concentrations at treatment entry on the one hand, and several biological and clinical parameters on the other hand. Interestingly, TMAO (Log transformed) correlated with various parameters such as α2 macroglobulin (*p* = 0.0007), glycated haemoglobin (*p* = 0.0008), total serum protein (*p* = 0.01), sodium (*p* = 0.009), ASAT/ALAT ratio (*p* = 0.031), glycemia (*p* = 0.037), Hepascore (*p* = 0.037), and length of hospital stay (*p* = 0.0067; see Appendix A). In addition, IS concentrations at treatment entry correlated with the AUDIT total score (*p* = 0.0032), platelets levels (*p* = 0.038), ASAT/ALAT ratio (*p* = 0.0465) and the mean number of self-reported meals per day before hospitalization (*p* = 0.0082, see Appendix A).

### 3.5. Correlations of Serum TMAO or IS Concentrations in AUD Patients with the Variation of Prealbumin or Albumin Concentrations during the Acute Phase of Alcohol Withdrawal

Serum albumin and prealbumin, two biological markers of nutritional status, were also measured at the end of the acute phase of alcohol withdrawal for 27 AUD patients, and it is of interest to note that variations of serum prealbumin during the acute phase of alcohol withdrawal correlated with IS concentration at the entry (Figure 2). In addition, variations of serum prealbumin correlated with variations of albumin during alcohol withdrawal (data not shown). Typically, prealbumin seems to decrease during alcohol withdrawal in most AUD patients, but that seems to be less important when AUD patients present medium to high levels of IS at treatment entry (Figure 2).

### 3.6. Predictors of TMAO and IS Concentrations in AUD Patients

We attempted to identify significant predictors of serum TMAO and IS concentrations at the entry by applying a backward stepwise multiple regression analyse. α2 macroglobulin (β = −0.497, *p* = 0.00069), ASAT/ALAT ratio (β = −0.454, *p* = 0.0013) and natremia (β = 0.423, *p* = 0.0025) were identified as significant predictors of serum TMAO concentrations (adjusted r^2^ = 0.7059, *p* = 0.0000127 for whole model; see Appendix A. The AUDIT score was the only predictor for IS concentrations (β = −0.517, *p* = 0.004058; see Appendix A.

### 3.7. Predictors of Clinical, Biological and Neuropsychological Parameters in AUD Patients

Finally, we also attempted to identify predictors for different clinical and neuropsychological parameters. Indicating the severity of AWS, the Cushman’s maximum score was best predicted by alkaline phosphatase (ALP) and carbohydrate deficient transferrin (CDT) (data not shown). No relationship was also shown between TMAO or IS concentrations and duration of benzodiazepin prescription, which is also an indicator of the severity of AWS (data not shown).

Concerning the neuropsychological assessment of AUD patients, age and glycemia were only the best predictors for BEARNI total score, which detect neuropsychological impairments in AUD patients. In this study, we failed to find significant predictors for MoCA (data not shown). When studying serum prealbumin variation during alcohol withdrawal, it is interesting to note that IS concentration was the best positive predictor (β = 0.408, *p* = 0.002363), and Albuminemia (β = −0.527, *p* = 0.00016) and Hyaluronic acid (β = −0.487, *p* = 0.000516) were the best negative predictor (adjusted r^2^ = 0.668, *p* = 0.00000533 for whole model; see Appendix A).

Finally, TMAO concentrations (Log transformed) at the entry was the only predictor of length of hospital stay in our population of AUD patients (β = −0.622, *p* = 0.001996; see Appendix A).

## 4. Discussion

Chronic alcohol consumption is often associated with gut and nutritional disorders, and AUD patients can develop brain damages leading to cognitive deficits. TMAO and IS are two metabolites coming from transformation of various dietary compounds by microbiota and then liver. Whereas there is growing interest for TMAO and IS in cardiovascular and neurological diseases, at our knowledge, no study assessed TMAO and IS concentrations in AUD patients. We thus aimed at measuring TMAO and IS concentrations in a group of AUD patients and at examining, in these patients, the relationships between serum levels on the one hand and biological, clinical or neuropsychological characteristics on the other hand.

In this study, we showed that serum TMAO concentrations in AUD patients presented an important interindividual variability when compared to HC. It is in agreement with literature data as the concentrations observed in our population were comparable to those reported in other studies [8]. Very low or very high TMAO concentrations were not observed in HC, potentially because the number of HC was limited in our study. As TMAO concentration may vary with dietary intake, liver function or dysbiosis, it would be then interesting to quantify these parameters in a larger cohort to assess the biological variability in nonpathological conditions.

We showed that many AUD patients had very high levels of serum TMAO, which may be deleterious. Indeed, deleterious effects of high plasma concentrations of TMAO and IS in the development of cardiovascular and/or neurological pathologies have been previously reported [20,24,30,31,32]. Regarding AUD patients, they have a higher risk to develop cardiovascular disease [33] such as hypertension, myocardial infarction or ischemic stroke. We can therefore hypothesis that TMAO or IS may play a role in the development of cardiovascular disease in AUD patients, but this hypothesis presupposes that high concentrations may be found in these patients, and that it could be deleterious. However, surprisingly, AUD patients with very high serum TMAO concentrations presented the weakest biological perturbations compared to other AUD patients. It is intriguing as high levels of TMAO are often presented in literature data as promoting deleterious effects contributing to diseases. One hypothesis is that the diet in this subgroup of patients may be close to a balanced Western diet (in quantity and quality), which would perhaps allow to have sufficient nutritional intakes of micronutrients (trace elements and vitamins) to fight against the harmful metabolic effects of alcohol (acetaldehyde production and oxidative stress). Another explanation could be the consumption of red meat in these patients the day before blood sampling during refeeding in the hospital setting.

Interestingly, many AUD patients had very low serum TMAO concentrations, and these patients presented unexpectedly more biological perturbations for different biomarkers assessing nutritional status and liver function. These effects are of interest but must be confirmed in a larger cohort of AUD patients.

We also report that AUD patients exhibit significant lower serum IS concentrations compared to HC. It should be noted that very high IS concentrations were not found in our group of AUD patients.

All these data were unexpected and different explanations can be proposed. Several factors probably affect the production of TMAO and IS: First, low concentrations of TMAO and IS could be related to a lower choline, betaine, carnitine or tryptophane dietary intake, which is dependent from the diversity, the quality and the quantity of food intake in a population of AUD patients which is often underprivileged. For different reasons, AUD patients can have reduced aminoacid and micronutrients intake, which directly affects TMA and indolic compounds production. A reduced food intake can further induce chronic denutrition and may promote a clinical state with a low-grade inflammation. In the present study, IS concentration at treatment entry, but not TMAO, was correlated with the number of meals per day in AUD patients, which reflects probably a lower tryptophan intake. Regarding TMAO, several studies showed an effect of diet but others also failed to find such an effect [8,13,34,35,36]. In the present study, AUD patients and HC have a Western diet and we cannot exclude an effect of such diet. Unfortunately, the nutritional intakes of choline, carnitine and betaine have not been investigated, and these intakes could not also be controlled in AUD patients during the days before blood sampling.

If we consider diet to be the main factor that affects TMAO or IS production, then our data suggest that medium or high serum levels of TMAO or IS could reflect adequate food intake and probably a good nutritional status in AUD patients at treatment entry, and this could be an indicator of a good prognosis in these patients.

Second, hepatic function is another factor that can explain low concentrations of TMAO or IS observed in some AUD patients. TMA and indolic compounds go through the gut and follow the veinous portal system up to the liver, and then these metabolites are transformed by hepatic enzymes such as FMO1 and FMO3 [11,35]. We can hypothesize that a reduced metabolic function of the liver affects the metabolism of TMA and indolic compounds, and reduces the levels of TMAO and IS concentrations in the blood of these patients. Then, low concentrations of TMAO or IS could reflect liver dysfunction, in agreement with the levels of different biomarkers assessing hepatic cytolysis (ASAT/ALAT ratio), cholestase and oxidative stress in the liver (γGT, α2 macroglobulin), or fibrosis (Hepascore).

Third, we can also hypothesize that alcohol consumption could modify composition of microbiota and alter the quantity of bacteria in the gut, as described in previous studies [37,38]. For example, it could modify TMA production in the gut and then TMAO concentrations in blood after hepatic metabolism. Alcohol withdrawal and renutrition can then promote changes of microbiota composition and contribute to variations of serum TMAO observed during withdrawal [38].

Finally, we did not show any link between serum concentrations of TMAO or IS, and the severity of AUD, the severity of AWS and cognitive deficits at the end of the acute phase of alcohol withdrawal. These findings suggest that the level of TMAO or IS does not affect directly neurological state of these AUD patients. Although we did not observe any effect in this study, we cannot exclude that an exposure to high concentrations of TMAO during several weeks or months is required to observe deleterious effects. In addition, although IS seems to be the main compound issued from tryptophan metabolism by microbiota, it should be noted that other indole derivatives, such as indol-3-acetic acid, can be produced and that they could also play a role in the development of liver, brain or cardiovascular diseases [3,39]. However, these compounds were not quantified in this study.

Considering the nutritional status, we showed in this study that IS level at treatment entry was a significant positive predictor of variation of serum prealbumin during alcohol withdrawal. Prealbumin is considered as a biomarker assessing denutrition and the efficiency of renutrition: an increase is observed in patients when renutrition is appropriated. In our study, serum prealbumin seems to decline during alcohol withdrawal in most AUD patients, probably reflecting metabolic changes during this period. However, the lower IS concentration at treatment entry, the more severe the decline of serum prealbumin concentration. These data encourage the identification of AUD patients with the lowest IS concentrations at treatment entry. They would require additional and specific nutritionist intervention associated with a personalized refeeding program and follow-up examinations during alcohol withdrawal.

Interestingly, we also showed that TMAO concentrations (Log transformed) is a significant predictor of the length of hospital stay. It is important to note that all AUD patients benefited from a personalized therapeutic course during their hospitalization. Therefore, the management of care may be heterogeneous between patients, some benefiting from specific cares such as education workshop, thus increasing the length of stay. It can induce bias in the interpretation of data and this relationship needs to be studied in a larger cohort.

Then, it could be of interest to further investigate a potential utility to quantify serum TMAO and IS in a larger cohort of AUD patients. It could be of importance to detect these patients early in the course of the treatment program in order to improve clinical practice, to improve nutritional status, and perhaps to reduce the length of hospital stay and the cost of hospitalization in this population of patients.

## 5. Strengths and Limitations of This Study

Several strengths could be mentioned in this work. First, the different inclusion criteria in this study allowed to have a homogeneous population of AUD patients, limiting potential biases. In addition, the use of modern analytical technologies allows to quantify more easily and specifically many metabolic compounds coming from microbiota and liver metabolism.

Nevertheless, this work has several limitations. The main limitation is the small sample size. Moreover, although AUD patients were strictly selected and our population can be considered as homogeneous, this population can also be considered as partially representative of AUD patients hospitalized in an addictology department. From a statistical point of view, we did not perform corrections for multiple comparisons on the correlations before entering in the regression model as this remains an exploratory analysis and we did not want to over-correct and take the risk of discarding true positives [40]. In addition, dietary intakes were not registered before the entry in the addiction care unit, and only few declarative data were available. The quantity and quality of the food ingested were not registered, and we cannot exclude that high concentrations of plasma TMAO observed in some AUD patients could result from the consumption of food from animal origin the day before blood sampling. At last, only IS was quantified in our patients, but not other metabolites from tryptophan. Thus, we cannot exclude that these other metabolites can have an effect on the severity of AWS or cognitive function during and after the hospitalization. In addition, metabolomic studies using Nuclear Magnetic Resonance could not be performed because these technologies were not available locally. This would have made it possible to identify and quantify many other metabolites in these samples and to better understand the mechanisms explaining low TMAO or IS concentrations observed in many AUD patients. Finally, neuropsychological tests were performed at the end of the acute alcohol withdrawal phase, and the effects of long-term exposure to high concentrations of TMAO or IS on cognitive function were not assessed. Therefore, we cannot exclude that long-term exposure to high levels of TMAO or IS is finally deleterious for AUD patients; it would require multiple blood sampling to assess evolution of these metabolites.

## 6. Conclusions

The present study is the first to explore TMAO and IS serum concentrations in AUD patients. Our data show first a relationship between low TMAO concentrations and higher biological perturbations, and second between low IS concentrations and the variations of serum prealbumin during the acute phase of alcohol withdrawal. In this population, AUD patients with high serum TMAO concentrations presented the weakest biological perturbations. Although we did not see any relationship between low TMAO or IS concentrations and the severity of AWS or cognitive deficits, our data suggest that AUD patients with low concentrations of TMAO or IS at treatment entry could be related to a poor nutritional status, and should probably benefit from a personalized refeeding program. Biological quantification of these metabolites in larger cohorts could be of interest in AUD patients.

## Figures and Tables

**Figure 1 nutrients-14-03964-f001:**
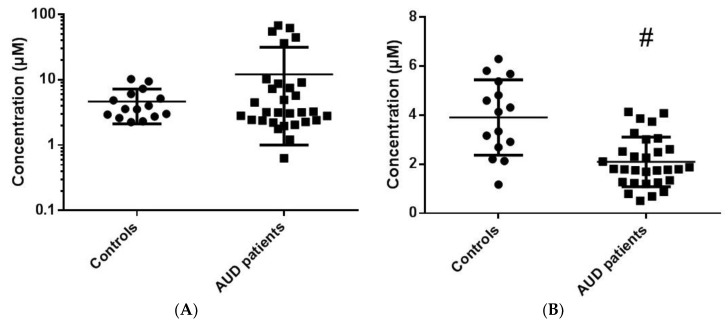
(**A**) TMAO and (**B**) IS concentrations in serum of AUD patients at the treatment entry and in HC. #: *p* < 0.05 when compared to controls. TMAO: Trimethylamine *N*-oxide. IS: Indoxyl Sulfate. AUD: Alcohol Use Disorder. HC: Healthy Controls.

**Figure 2 nutrients-14-03964-f002:**
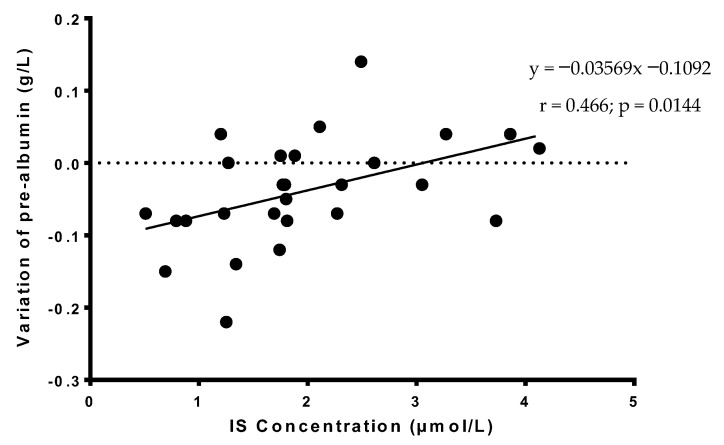
Correlation between serum IS concentrations of AUD patients at the treatment entry and variations of serum prealbumin concentrations during alcohol withdrawal. Statistically significant at *p* < 0.05.

**Table 1 nutrients-14-03964-t001:** Demographic, clinical, neuropsychological and biological data observed in AUD patients and HC. Data are expressed as the mean ± standard deviation. AUD: Alcohol Use Disorder. HC: Healthy Controls. AUDIT: Alcohol Used Disorders Identification Test. MoCA: Montreal Cognitive Assessment. BEARNI: Brief Evaluation of Alcohol-Related Neuropsychological Impairments. Significantly different at *p* < 0.05. NA: Not applicable. ND: Not done.

	AUD Patients (*n* = 30)	Healthy Controls (*n* = 15)	*p*
**Demographic and Clinical variables**			
Age (year)	47.5 ± 10.1	46.2 ± 5.1	NS
Sexe (M/F)	24/6	12/3	NS
Body Mass Index (kg/m^2^)	24.1 ± 4.3	24.3 ± 4.3	NS
AUDIT	28.6 ± 7.8	2.6 ± 1.4	*p* < 0.05
Cushman’s score	4.9 ± 2.2	NA	ND
Benzodiazepin prescription (days)	6.7 ± 5.3	NA	ND
MoCA score (/30)	24.5 ± 5.0	27.4 ± 1.4	*p* < 0.05
BEARNI score (/30)	13.5 ± 6.1	19.1 ± 2.9	*p* < 0.05
Length of hospital stay (days)	20.2 ± 5.6	NA	ND
**Biological measures**			
Glycemia (mmol/L)	5.3 ± 0.9	5.3 ± 0.6	NS
Glycated haemoglobin (%)	5.4 ± 0.6	5.4 ± 0.3	NS
Total serum protein (g/L)	66.4 ± 6.9	70.3 ± 3.5	*p* = 0.05
Albumin (g/L)	37.2 ± 4.7	43.2 ± 2.8	*p* < 0.05
Prealbumin (g/L)	0.27 ± 0.08	0.31 ± 0.04	*p* < 0.05
γ-glutamyl-transpeptidase (U/L)	442.2 ± 812.2	16.9 ± 10.5	*p* < 0.05
Ammonium (µmol/L)	42.8 ± 23.3	28.1 ± 7.2	*p* < 0.05
ASAT (U/L)	69.3 ± 60.3	19.6 ± 7.8	*p* < 0.05
ALAT (U/L)	50.3 ± 71,7	21.1 ± 12.5	*p* < 0.05
ASAT/ALAT ratio	2.2 ± 2.5	1.0 ± 0.3	*p* < 0.05
Bilirubin (µmol/L)	25.1 ± 17.4	17.9 ± 10.5	*p* < 0.05
α2 macroglobulin (g/L)	2.1 ± 0.6	1.7 ± 0.5	*p* < 0.05
Hyaluronic acid (µg/L)	121.4 ± 320.7	15.2 ± 5.5	*p* < 0.05
Score of fibrosis HEPASCORE	0.46 ± 0.31	0.22 ± 0.13	*p* < 0.05
Serum creatinine (µmol/L)	61.9 ± 12.7	78.7 ± 12.8	*p* < 0.05
Urea (mmoles/L)	2.7 ± 1.0	4.4 ± 0.8	*p* < 0.05
Haemoglobin (g/dL)	14.1 ± 1.4	14.8 ± 0.7	*p* < 0.05
Mean Cellular Volume (µm^3^)	98.0 ± 6.0	87.7 ± 4.3	*p* < 0.05
Platelets (G/L)	205 ± 82	280 ± 61	*p* < 0.05
Prothrombin time (%)	96.3 ± 8.4	96.9 ± 5.2	NS
Trimethylamine *N*-oxide TMAO (µmol/L)	12.1 ± 19.2	4.6 ± 2.5	NS
Indoxyl sulfate IS (µmol/L)	2.1 ± 1.0	3.9 ± 1.5	*p* < 0.05

## Data Availability

Not applicable.

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
