# Peer review of "Trimethylamine N-Oxide (TMAO) and Indoxyl Sulfate Concentrations in Patients with Alcohol Use Disorder"

_nutrients, 2022, doi:10.3390/nu14193964_

Round 1
Reviewer 1 Report
1. Introduction is very well written; it is informative and concise with a clear literature review choice and step-by-step approach.
2. Methods: were there any power calculation analyses? What about sample size calculation? Why did you include 30 and not, for example, 25, 35, 40, or 50 patients?
3. Results: In my opinion, the sample size is too small to show that in healthy controls there were no very low or very high TMAO concentrations (line 175-176)
What type of food did they have? Changing the type of food might affect the concentrations of both biomarkers.
4. Discussion: line 291: how do you explain the fact that high levels of TMAO are associated with weakest biological perturbations?
Author Response
RESPONSES TO REVIEWER 1
- Introduction is very well written; it is informative and concise with a clear literature review choice and step-by-step approach.
- Methods: were there any power calculation analyses? What about sample size calculation? Why did you include 30 and not, for example, 25, 35, 40, or 50 patients?
This study was an ancillary investigation; the two parameters studied herein were additional to the initial study. The number of patients is related to the sample size required to answer our main initial objective. It corresponds to patients with full data (biological, clinical, neuropsychological and imagery data). Another point is that patients and healthy controls included in this study were recruited to match the demographics (age, sex and education) of patients, as indicated in Material and methods.
In addition, the biological variability was not known in our population at the start of the study, and it was not possible to perform power calculation.
Because the initial study started in 2012, biological samples were still available for only 30 AUD patients and 15 controls. But we agree with the reviewer and it could be of interest to reproduce these results in a larger cohort. Moreover, further studies should explore a putative interest of the quantification of TMAO and/or IS in other clinical situations.
The beginning of paragraph 2.6 has been modified as follows (in red):
“2.6. Statistical analysis
This study was retrospective, as an ancillary investigation of large research protocol, and biological variability of IS or TMAO was not known in our population at the start of the study. It was then not possible to perform power calculation and to determine the number of subjects a priori.
All data were expressed as mean±standard deviation.”
- Results: In my opinion, the sample size is too small to show that in healthy controls there were no very low or very high TMAO concentrations (line 175-176)
We agree with the reviewer and we think that such low or high TMAO concentrations should be found in the general population, as TMAO concentration may vary with dietary intake, liver function or dysbiosis. It would then be interesting to quantify these parameters in a larger cohort to assess the biological variability in nonpathological conditions. Unfortunately, the number of controls subjects was limited in our study. Nevertheless, we believe that these data may be representative
In our opinion, and given the results that we observed, it may be more interesting to look for patients with a low concentration of TMAO (or IS) and to study a potential link with a clinical state (malnutrition?), not just in the context of AUD.
The discussion has been modified as follows (in red):
In this study, we showed that serum TMAO concentrations in AUD patients presented an important interindividual variability when compared to HC. It is in agreement with literature data as the concentrations observed in our population were comparable to those reported in other studies [8]. Very low or very high TMAO concentrations were not observed in HC, potentially because the number of HC was limited in our study. As TMAO concentration may vary with dietary intake, liver function or dysbiosis, it would be then interesting to quantify these parameters in a larger cohort to assess the biological variability in nonpathological conditions.
We showed that many AUD patients had very high levels of serum TMAO, which may be deleterious. Indeed, deleterious effects of high plasma concentrations of TMAO and IS in the development of cardiovascular and/or neurological pathologies have been previously reported [20,24,30–32]. Regarding AUD patients, they have a higher risk to develop cardiovascular disease [33] such as hypertension, myocardial infarction or ischemic stroke. We can therefore hypothesis that TMAO or IS may play a role in the development of cardiovascular disease in AUD patients, but this hypothesis presupposes that high concentrations may be found in these patients, and that it could be deleterious. However, surprisingly, AUD patients with very high serum TMAO concentrations presented the weakest biological perturbations compared to other AUD patients. It is intriguing as high levels of TMAO are often presented in literature data as promoting deleterious effects contributing to diseases.
What type of food did they have? Changing the type of food might affect the concentrations of both biomarkers.
We can consider that these participants have a Western diet, but patients eat very little; the number of meals/day is a declarative data and should be taken with caution, because of confusion due to alcohol consumption or memory deficit.
The discussion has been modified as follows (in red):
Regarding TMAO, several studies showed an effect of diet but others also failed to find such an effect [8,13,34–36]. In the present study, AUD patients and HC have a western diet and we cannot exclude an effect of such diet. Unfortunately, the nutritional intakes of choline, carnitine and betaine have not been investigated, and these intakes could not also be controlled in AUD patients during the days before blood sampling.
- Discussion: line 291: how do you explain the fact that high levels of TMAO are associated with weakest biological perturbations?
These data are intriguing and the discussion has been modified as follows (in red):
We showed that many AUD patients had very high levels of serum TMAO, which may be deleterious. Indeed, deleterious effects of high plasma concentrations of TMAO and IS in the development of cardiovascular and/or neurological pathologies have been previously reported [20,24,30–32]. Regarding AUD patients, they have a higher risk to develop cardiovascular disease [33] such as hypertension, myocardial infarction or ischemic stroke. We can therefore hypothesis that TMAO or IS may play a role in the development of cardiovascular disease in AUD patients, but this hypothesis presupposes that high concentrations may be found in these patients, and that it could be deleterious. However, surprisingly, AUD patients with very high serum TMAO concentrations presented the weakest biological perturbations compared to other AUD patients. It is intriguing as high levels of TMAO are often presented in literature data as promoting deleterious effects contributing to diseases. One hypothesis is that the diet in this subgroup of patients may be close to a balanced western diet (in quantity and quality), which would perhaps allow to have sufficient nutritional intakes of micronutrients (trace elements and vitamins) to fight against the harmful metabolic effects of alcohol (acetaldehyde production and oxidative stress). Another explanation could be the consumption of red meat in these patients the day before blood sampling during refeeding in the hospital setting.
Interestingly, many AUD patients had very low serum TMAO concentrations, and these patients presented unexpectedly more biological perturbations for different biomarkers assessing nutritional status and liver function. These effects are of interest but must be confirmed in a larger cohort of AUD patients.

Reviewer 2 Report
This paper aims to examine serum trimethylamine N-oxide (TMAO) and indoxyl sulfate (IS) concentrations in patients with alcohol-use disorder (AUD) at the entry for alcohol withdrawal, and the relationships with several biological, neuropsychological, and clinical parameters. For that purpose, authors have quantified TMAO and IS levels in thirty AUD inpatients and fifteen healthy controls (HC). The results presented showed a relationship between low serum TMAO levels in AUD patients and higher biological perturbations. In addition, the results also prove a relationship between low IS concentrations and the variations of serum prealbumin during the acute phase of alcohol withdrawal.
The article is the result of a well-planned study at the beginning with some logical exclusion criteria although with a small number of patients which weakens the robustness of the conclusions obtained. The discussion of the results has been carried out correctly, with simple but direct figures. The conclusions are supported by the results. The hypotheses included that explain part of the complexity of the results obtained agree with the results and the bibliography and are not excessively speculative. Finally, the text presents some conclusions that I consider may be interesting for present and future work.
Next, I would like to comment on certain points of the work that I have doubts or that I consider could be improved:
- In the discussion section I have found the statements contained in the lines 303-308 a bit contradictory with the data collected in table S1 and S2. The authors claim: low concentrations of TMAO and IS could be related to a lower dietary intake in AUD patients, but the BMI and number of meals per day of patients included in the first and third tertile according their TMAO or IS levels are very similar.
- The choice of mass spectrometry as an analytical technique for the determination of TMAO and IS levels in serum samples was correct but incomplete. As the authors indicate in the introduction to the article, the determination of other common serum metabolites such as choline, betaine, L-carnitine or even certain lipids would have been of great interest and probably key to understanding the altered mechanisms. The study of metabolites related, for example, to kidney failure, such as creatine/creatinine levels, or deteriorated liver function (alcoholic cirrhosis) would notably enrich the results of the study. The discussion of the results obtained is a bit poor without being able to include these compounds in it. The determination of the concentration of those metabolites, including TMAO and IS, could have been carried out simply, quickly and at a lower economic cost using an NMR metabolomic platform.
Author Response
RESPONSES TO REVIEWER 2
Next, I would like to comment on certain points of the work that I have doubts or that I consider could be improved:
1- In the discussion section I have found the statements contained in the lines 303-308 a bit contradictory with the data collected in table S1 and S2. The authors claim: low concentrations of TMAO and IS could be related to a lower dietary intake in AUD patients, but the BMI and number of meals per day of patients included in the first and third tertile according their TMAO or IS levels are very similar.
In this study, the number of meals per day is a declarative data and should be taken with caution, because AUD patients may have memory deficits or confusion due to alcohol consumption. Then, AUD patients’ diet may vary in terms of quality or quantity of food, and a lower dietary intake may be an explanation of such low concentrations of IS or TMAO.
In addition, population of AUD patients is often underprivileged and can have no access to meat or food from animal origin, perhaps limiting intakes of choline, betaine, L-carnitine and tryptophane which are precursors of TMAO or IS, and explaining the absence of difference of BMI between subgroups.
The discussion has been modified as follows (in red):
All these data were unexpected and different explanations can be proposed. Several factors probably affect the production of TMAO and IS: First, low concentrations of TMAO and IS could be related to a lower choline, betaine, carnitine or tryptophane dietary intake, which is dependent from the diversity, the quality and the quantity of food intake in a population of AUD patients which is often underprivileged. For different reasons, AUD patients can have reduced aminoacid and micronutrients intake, which directly affects TMA and indolic compounds production.
2- The choice of mass spectrometry as an analytical technique for the determination of TMAO and IS levels in serum samples was correct but incomplete. As the authors indicate in the introduction to the article, the determination of other common serum metabolites such as choline, betaine, L-carnitine or even certain lipids would have been of great interest and probably key to understanding the altered mechanisms. The study of metabolites related, for example, to kidney failure, such as creatine/creatinine levels, or deteriorated liver function (alcoholic cirrhosis) would notably enrich the results of the study. The discussion of the results obtained is a bit poor without being able to include these compounds in it. The determination of the concentration of those metabolites, including TMAO and IS, could have been carried out simply, quickly and at a lower economic cost using an NMR metabolomic platform.
We agree with the reviewer 2.
NMR is currently extensively used in metabolomic studies, and it is a tool likely to develop more and more in the future. We agree that many other metabolites (such as TMA or Tryptophane and others) could be studied using a single sample. For example, quantification of TMA may help to clarify the role of the liver in the clearance of TMA in a context of AUD, and if low concentrations of TMAO observed in some AUD patients could be the result of an altered liver function. Unfortunately, this technology was not available in our hospital, that why LCMS was our method of choice in this study.
Regarding this study, we aimed also at developing methods presenting good analytical performance (in terms of limit of quantification, repeatability, reproducibility, and so on..) to meet the standard of ISO15189 certification and with the perspective of a potential use for clinical applications.
LCMSMS is currently the gold standard to quantify metabolites in blood or urine samples using MRM transitions, and it seemed more appropriate for us to employ known and mastered methods usable later for routine clinical applications.
The strengths and limitation paragraph has been modified as follows (in red):
- Strengths and limitations of this study
Several strengths could be mentioned in this work. First, the different inclusion criteria in this study allowed to have a homogeneous population of AUD patients, limiting potential biases. In addition, the use of modern analytical technologies allows to quantify more easily and specifically many metabolic compounds coming from microbiota and liver metabolism.
Nevertheless, this work has several limitations. The main limitation is the small sample size. Moreover, although AUD patients were strictly selected and our population can be considered as homogeneous, this population can also be considered as partially representative of AUD patients hospitalized in an addictology department. From a statistical point of view, we did not perform corrections for multiple comparisons on the correlations before entering in the regression model as this remains an exploratory analysis and we did not want to over-correct and take the risk of discarding true positives [40]. In addition, dietary intakes were not registered before the entry in the addiction care unit, and only few declarative data were available. The quantity and quality of the food ingested were not registered, and we cannot exclude that high concentrations of plasma TMAO observed in some AUD patients could result from the consumption of food from animal origin the day before blood sampling. At last, only IS was quantified in our patients, but not other metabolites from tryptophan. Thus, we cannot exclude that these other metabolites can have an effect on the severity of AWS or cognitive function during and after the hospitalization. In addition, metabolomic studies using Nuclear Magnetic Resonance could not be performed because this technology was not available locally. This would have made possible to identify and quantify many other metabolites in these samples and to better understand the mechanisms explaining low TMAO or IS concentrations observed in many AUD patients. Finally, neuropsychological tests were performed at the end of the acute alcohol withdrawal phase, and the effects of long-term exposure to high concentrations of TMAO or IS on cognitive function were not assessed. Therefore, we cannot exclude that long-term exposure to high levels of TMAO or IS is finally deleterious for AUD patients; it would require multiple blood sampling to assess evolution of these metabolites.

Round 2
Reviewer 1 Report
Thank you for providing all the answers to questions and comments. I congratulate the authors for preparing an interesting and informing manuscript.